# Broad Host Range Peptide Nucleic Acids Prevent Gram-Negative Biofilms Implicated in Catheter-Associated Urinary Tract Infections

**DOI:** 10.3390/microorganisms13081948

**Published:** 2025-08-20

**Authors:** Hannah Q. Karp, Elizabeth S. Nowak, Gillian A. Kropp, Nihan A. Col, Michael D. Schulz, Nammalwar Sriranganathan, Jayasimha Rao

**Affiliations:** 1Virginia Tech Carilion School of Medicine, Roanoke, VA 24016, USA; hqkarp@vt.edu (H.Q.K.); esnowak@carilionclinic.org (E.S.N.); 2Carilion Clinic Department of Infectious Disease, Carilion Clinic, Roanoke, VA 24014, USA; 3Virginia Tech Department of Chemistry, Virginia Tech, Blacksburg, VA 24061, USA; 4Department of Biomedical Sciences and Pathobiology, Virginia-Maryland College of Veterinary Medicine, Virginia Tech, Blacksburg, VA 24061, USAnathans@vt.edu (N.S.)

**Keywords:** biofilm, peptide nucleic acids, CAUTIs, novel agents

## Abstract

Biofilms develop in sequential steps resulting in the formation of three-dimensional communities of microorganisms that are encased in self-produced extracellular polymeric substances. Biofilms play a key role in device-associated infections, such as catheter-associated urinary tract infections (CAUTIs), because they protect microorganisms from standard antimicrobial therapies. Current strategies to prevent biofilm formation in catheter-related infections, including prophylactic antibiotics and antibiotic-coated catheters, have been unsuccessful. This finding highlights a need for novel approaches to address this clinical problem. In this study, biofilm-forming phenotypes of common Gram-negative bacteria associated with CAUTIs were treated with antisense peptide nucleic acids (PNAs), and biofilm biomass and bacterial viability were quantified after 24 h of treatment. A cocktail of PNAs targeting the global regulator genes *rsmA*, *amrZ*, and *rpoS* in *Pseudomonas aeruginosa* significantly reduced viability and thus appropriately eliminated biofilm biomass. Antisense-PNAs against these same gene targets and the motility regulator gene *motA* inhibited biofilm formation among isolates of *Klebsiella pneumoniae*, *Enterobacter cloacae*, and *Escherichia coli* but did not reduce bacterial viability. These results suggest that antisense-PNAs are a promising new technology in preventing biofilm formation in urinary catheters, especially as a potential complement to conventional antimicrobials.

## 1. Introduction

Bacterial biofilms in the healthcare setting are critical to address because they pose a risk to patient health. It is estimated that more than 65% of nosocomial infections are due to biofilms, which are implicated in surgical site infections, bloodstream infections, and catheter-associated urinary tract infections (CAUTIs) [1]. Up to 25% of hospitalized patients receive a urinary catheter, and roughly 75% of urinary tract infections developed in the hospital are associated with catheter use [2]. Biofilms form on medical devices and can confer resistance to antimicrobials and host immune defenses [3]. This resistance may arise by several proposed mechanisms, including conventional resistance mechanisms, the protective barrier formed by biofilms, and the multicellular nature of biofilms, which involves inter-bacterial signaling via quorum sensing (QS) [3,4].

Gram-negative bacilli are the leading cause of CAUTIs, including *Pseudomonas aeruginosa*, *Escherichia coli*, select *Klebsiella* spp., and select *Enterobacter* spp. [5,6]. Clinically, best practices related to urinary catheters include minimizing the frequency and duration of their use, and studies suggest antimicrobial prophylaxis in asymptomatic patients is not beneficial [7]. Attempts to address biofilms in the pathogenesis of CAUTIs include the use of antifouling and antibacterial surfaces leveraging polyethylene glycol or metal ions, silver alloy coated catheters, interrupting QS via degradation of signal molecules and QS inhibitors, and nanoparticle delivery of antibiotics [3,4,8,9]. These technological approaches have been insufficient in clinical use and challenged by multiantibiotic resistance, thus highlighting the need for novel methods to prevent bacterial biofilm formation in urinary catheters [10].

Synthetic antisense peptide nucleic acids (PNAs) can be harnessed to inhibit gene expression, and their use in prokaryotic systems was demonstrated in a biofilm-forming laboratory strain of *P. aeruginosa* [11,12,13]. PNAs are synthetic DNA analogs created by replacing the negatively charged phosphate backbone of DNA with a neutral pseudopeptide backbone, with nitrogenous bases attached by methylene carbonyl bonds [13,14,15]. PNAs form DNA and RNA hybrids, abiding by Watson–Crick–Franklin hybridization rules, with high specificity and selectivity. The resultant PNA-DNA and PNA-RNA complexes are more stable than the natural complexes since there is no electrostatic repulsion [13,15,16]. Additional properties of PNAs include high thermal stability, hybridization at almost any salt concentration, and resistance to proteases and nucleases [13,16]. The desired PNA sequence is addended to a cell-wall permeabilizing peptide (CPP) to improve entry of the PNA through cell membranes and an O-linker to improve solubility [11,17]. While PNAs bind to targets with high specificity, this characteristic may limit their effectiveness against a range of bacteria or polymicrobial infections.

In this study, novel antisense-PNAs were designed to prevent attachment of bacteria to a surface, which is a critical component of the first two steps out of five that describe biofilm formation [3]. These antisense-PNAs prevented biofilm formation and reduced bacterial viability in a biofilm-forming laboratory strain of *P. aeruginosa*. Interestingly, many of the key regulatory genes of biofilm formation are relatively conserved across Gram-negative bacilli implicated in CAUTIs. Therefore, antisense-PNAs were designed to capitalize on these shared areas of gene sequences, targeting global regulators of biofilm among *Escherichia coli*, *Klebsiella pneumoniae*, and *Enterobacter cloacae*.

## 2. Materials and Methods

### 2.1. Bacterial Strains

A laboratory strain of *P. aeruginosa*, PAO1 (Nottingham-subline), biofilm-forming clinical isolates of *K. pneumoniae* and *E. coli*, and a laboratory strain of *E. cloacae* (ATCC 13047, Manassas, VA, USA) were utilized (Appendix A, Figure A1). PAO1 was a kind gift from Dr. Stephan Heeb (University of Nottingham, Nottingham, UK) [18]. A total of 25–30% (vol/vol) glycerol stocks of bacteria were streaked onto Tryptic Soy Agar (TSA, Millipore Sigma, Burlington, MA, USA) or MacConkey Agar (MAC, Millipore Sigma) plates and incubated overnight at 37 °C. Single colonies were then subcultured in 4 mL of Luria Bertani (LB) Miller’s media (Thermo Fisher, Waltham, MA, USA) overnight at 37 °C in shaking conditions at 200 revolutions per minute (rpm) for 18 h, after which they were normalized to 0.01 = OD_600_.

### 2.2. Antisense-PNA Design and Synthesis

Antisense-PNAs were designed using NCBI BLAST+ 2.17.0 to select a 12 to 14 base pair region inclusive of the start codon. The start codon was included because the PNAs were designed to prevent transcription of the target gene. The PNA Tool (PNA Bio) was used to provide suggestions on optimal design to maximize solubility, which included suggestions such as a G content ≤ 35% [19]. The 12 to 14 base pair range was determined from the previous literature reports that suggested at least a 10 base pair length of PNA is needed to form stable PNA-DNA hybrids, while at the same time the PNA needs to be long enough to bind to a targeted area of a specific gene [13]. Shorter PNAs also have increased cellular uptake and internalization by bacteria compared to longer PNAs [20].

PNA Bio (Newberry Park, CA, USA) created these antisense-PNAs via linear synthesis of PNA, CPP, and O-linker on resin. Each sequence at the 5′ end was conjugated to a 9-atom organic compound (O-linker) to improve solubility via an increase in hydrophilicity and the CPP (KFF)_3_K to improve cell wall permeability [11,21]. The mode of PNA delivery with the CPP (KFF)_3_K is believed to be via the inner membrane ABC transporters [20]. All antisense-PNAs were synthesized and assessed for purity by PNA Bio and then shipped as a dry powder for reconstitution in the laboratory.

### 2.3. Antisense-PNA Treatment

Bacterial subcultures, as described in Section 2.1 of methods were normalized to optical density (OD) 600 = 0.01 in minimal M9+ media (supplemented with sterile 20% glucose, 1 M MgSO_4_, 0.1 mM CaCl_2_, 3% casamino acid, and hydroxyquinoline solution). These supplements, previously described in Louden et al., functioned to enhance biofilm formation by slowing growth and altering protein synthesis since 8-hydroxyquinolone acts as an iron chelator [22]. Standard 10-fold serial dilutions were made utilizing M9+ supplemented media (Thermo Fisher), except N minimal media was utilized for *E. cloacae* [23]. The standardized subcultures contained roughly 1.7 × 10^7^ colony-forming units (CFU)/mL, which was verified by serial dilution and quantification of CFU/mL in triplicates. For a given 96-well clear flat-bottom polystyrene tissue culture-treated microplate (herein referred to as a 96-well plate), the final volume was 60 µL and consisted of the following unless otherwise stated: 10 µM per antisense-PNA, 30 µL of 1.7 × 10^7^ CFU/mL bacterial solution, and the remaining volume was sterile deionized water. Each condition was performed in triplicate. The plate was incubated statically at 30 °C for 24 h in a humid chamber. The biofilm-forming laboratory strain of *P. aeruginosa*, PAO1, was used as a positive control. A total of 10 µM per antisense-PNA was based on the literature exploring what concentration of PNA is required to inhibit bacterial growth [17].

For polymicrobial biofilm, the 30 µL of normalized bacteria described above consisted of 30 µL total of equal parts of 1.7 × 10^7^ CFU/mL of PAO1 and 1.7 × 10^7^ CFU/mL of a second bacterial species.

### 2.4. Drying a Cocktail of PNAs onto 96-Well Plates

An equimolar (10 µM per antisense-PNA) cocktail of the antisense-PNAs *rsmA* 0, *amrZ* 0, and *rpoS* 0 (termed cocktail of PNAs, or cPNA) with 50% methanol (MeOH) (total volume: 60 µL) was dried onto a 96-well plate in a Fisher brand™ Isotemp™ Digital Dry Bath/Block Heater (Waltham, MA, USA) at 37 °C. A total of 60 µL of 100 µg/mL gentamicin was also dried onto the wells. Gentamicin-resistant *P. aeruginosa*, *Pa383*-∆ *rahU*::GM was subcultured as described in Section 2.1 [24].

### 2.5. Bacterial Viability Assessments

Direct stamping, using a 48-pin micro-plate pin replicator, on TSA or MAC plates was performed first after antisense-PNA treatment was complete. Then, 10-fold serial dilutions of each well were performed in LB media. A total of 30 µL of each dilution, in duplicate, was streaked on TSA or MAC plates to determine CFU/mL. The plates were incubated overnight at 37 °C and then were evaluated by ≥2 readers.

For polymicrobial samples, the serial dilutions were plated on TSA and MAC plates to highlight phenotypic differences.

### 2.6. Biofilm Biomass Quantification

Biofilms were stained with 175 µL 0.1% crystal violet (CV), rinsed with sterile water until the water ran clear, solubilized in 100 µL of 33% glacial acetic acid, and absorbance was measured using a spectrophotometer at 590 nm (Biomate 5 Spectrophotometer, Thermo Fisher, Waltham, MA, USA) [25].

### 2.7. Statistical Analyses

Results are presented as mean ± standard deviation (s.d.). *p* values were determined using unpaired *t*-tests (α = 0.05), one-way analysis of variance (ANOVA) with follow-up tests comparing the mean of each column to the control column (α = 0.05), or two-way ANOVA with multiple comparisons comparing cell means with others in its row and its column (family-wise α threshold and confidence level = 0.05). All graphs and statistics were created and performed in GraphPad Prism 10.

## 3. Results

### 3.1. PNA Design

The PNAs in this study were designed to target global regulator genes critical to the formation and maintenance of biofilms. The selected gene targets included *rsmA* (a translation regulator including proteins involved in QS), *amrZ* (a transcription factor that regulates twitching motility and alginate synthesis), and *rpoS* (a stress response and biofilm architecture regulator) [26,27,28]. The flagellar motility gene *motA* is also recognized as a critical mediator of early biofilm establishment through its role in polar reversible attachment between bacteria and a surface as well as additional adherence events during the irreversible phase of biofilm development [3,11]. Each PNA was labeled according to its intended gene target. The original PNAs were designed based on the PAO1 genome and are designated as “0”. Subsequent PNAs with homologous targets were labeled sequentially (Table 1).

### 3.2. A Cocktail of PNAs Inhibited Biofilm Formation in PAO1 and Reduced Bacterial Viability

Prior studies demonstrated inconsistent effects of single antisense-PNA treatment on PAO1 biofilm formation and bacterial viability (Appendix B, Figure A2). Therefore, combination treatment with multiple PNA targets was pursued. A 24 h treatment with an equimolar cocktail of the three antisense-PNAs *rsmA* 0, *amrZ* 0, and *rpoS* 0 (cPNA) eliminated biofilm formation in PAO1 (*p* < 0.001) and significantly reduced average bacterial viability by a fold change of 0.043 (*p* < 0.001) (Figure 1a,b). A 24 h treatment with *motA* 0 antisense-PNA eliminated biofilm formation (*p* < 0.001) and significantly reduced average bacterial viability by a fold change of 0.289 (*p* < 0.001) (Figure 1c,d). In summary, the data demonstrated that treatment with a cPNA essentially eliminated bacterial viability in PAO1 and thus biofilm biomass, and treatment with *motA* 0 antisense-PNA eliminated biofilm formation and reduced bacterial viability in PAO1.

### 3.3. Antisense-PNAs Inhibited Biofilm Formation in Their Intended Species with No Meaningful Reduction in Bacterial Viability

Biofilm formation by clinical isolates of *K. pneumoniae*, *E. cloacae*, and *E. coli* was quantified. A biofilm-forming phenotype was qualified as an average biofilm biomass ≥ 0.1. Biofilm-forming clinical isolates were identified for *K. pneumoniae* and *E. coli* (Figure A1). No biofilm-forming phenotype was identified among the *E. cloacae* clinical isolates screened, so a commercially purchased strain (ATTC 13047) was utilized (Figure A1).

Antisense-PNA targets were designed for bacterial species commonly implicated in CAUTIs (Table 1). A 24 h antisense-PNA treatment with *rsmA* 1 eliminated biofilm biomass in *K. pneumoniae* (*p* < 0.005) with no meaningful reduction in bacterial viability (Figure 2a,b). A 24 h antisense-PNA treatment with an equimolar cocktail of *rsmA* 1 and *amrZ* 2 significantly reduced average biofilm biomass in commercial *E. cloacae* strain ATCC 13047 by a fold change of 0.436 (*p* < 0.01), and there was no meaningful reduction in bacterial viability (Figure 2c,d). A 24 h antisense-PNA treatment with an equimolar cocktail of *rsmA* 1 and *amrZ* 1 eliminated biofilm biomass in two of the three clinical isolates of *E. coli* (*p* < 0.005), but insignificantly reduced average biofilm biomass in *E. coli* isolate three (*p* = 0.054) (Figure 2e). There was no meaningful reduction in average bacterial viability with 24 h antisense-PNA treatment of *E. coli* (Figure 2f). Overall, the novel antisense-PNAs designed for common causative organisms of CAUTIs demonstrated significant reduction in biofilm biomass in their intended species upon treatment [5].

### 3.4. Antisense-PNAs Reduced Biofilm Biomass in Polymicrobial Samples

*K. pneumoniae*, a lactose-fermenter, was cultured at a 1:1 ratio with PAO1, a non-lactose fermenter, and this was confirmed morphologically (Figure 3a). Upon 24 h treatment with cPNA, average biofilm biomass was significantly reduced by a log change of 0.382 (*p* < 0.001), PAO1 bacterial viability was eliminated, and *K. pneumoniae* bacterial viability was unchanged compared to the untreated control (Figure 3b,c). Upon switching the *rsmA* 0 antisense-PNA in the cPNA with *rsmA* 1 antisense-PNA, the reduction in biofilm biomass was lost (*p* = 0.726); however, PAO1 viability was still eliminated (Figure 3b,c). When the polymicrobial sample was treated with antisense-PNA *rsmA* 1 only, there was still a significant reduction in average biofilm biomass compared to the untreated control with a log change of 0.482 (*p* < 0.001), and PAO1 bacterial viability was still eliminated (Figure 3b,c). Regardless of whether the polymicrobial sample was treated with an equimolar cocktail of *rsmA* 1, *amrZ* 0, and *rpoS* 0 or *rsmA* 1 only, there was no favorable reduction in *K. pneumoniae* bacterial viability (Figure 3c). These results suggest that the cPNA retained their anti-biofilm effects even in more complex samples.

### 3.5. Dried Antisense-PNAs Retained Their Anti-Biofilm Effect

A 24 h incubation with cPNA eliminated bacterial viability of gentamicin-resistant *P. aeruginosa*, *Pa383*-∆*rahU*::GM, regardless of whether the cPNA was in solution (aq) or dried onto a 96-well plate with methanol (MeOH) (Figure 4a). Average biofilm biomass after 32 h incubation was not significantly different between untreated *Pa383*-∆*rahU*::GM in standard M9+ media and in 50% MeOH dried control wells (*p* = 0.622) (Figure 4b). There were no statistically significant differences between the average biofilm biomass of *Pa383*-∆*rahU*::GM treated with gentamicin (aq) and gentamicin dried in MeOH (*p* = 0.463), untreated bacteria in M9+ media and gentamicin (aq) (*p* = 0.242), and untreated bacteria in dried MeOH control wells and gentamicin dried in MeOH (*p* = 0.161) (Figure 4b). A total of 32 h incubation with cPNA eliminated biofilm biomass in the aq and dried forms (Figure 4b). Regardless of if the cPNA were in an aq or dried form, they retained their anti-biofilm effect and bactericidal effect against gentamicin-resistant *P. aeruginosa*, *Pa383*-∆*rahU*::GM.

## 4. Discussion

CAUTIs adversely impact patient health and the financial well-being of healthcare systems. Bacteria within biofilms are 10 to 1000 times more resistant to antibiotics than planktonic bacteria, which confers serious implications in the management of microbial infections secondary to bacterial biofilms [4]. Additionally, in the United States alone, the total economic burden of CAUTIs was estimated to be USD 1.7 billion annually (in 2016 dollars), highlighting both clinical and economic motivations to reduce the incidence of CAUTIs [29].

In this study, we demonstrated that antisense-PNAs are a promising solution to prevent biofilm establishment among many Gram-negative bacteria associated with CAUTIs. Inspired by the combination therapies used in a variety of conditions, such as human immunodeficiency virus, we investigated treating PAO1 with a cocktail of equimolar *rsmA* 0, *amrZ* 0, and *rpoS* 0 (cPNA), which eliminated biofilm formation and bacterial viability, suggesting a beneficial effect of simultaneously targeting multiple genes involved in biofilm formation and maintenance (Figure 1a,b) [30]. Assessment of viable but non-culturable cells was not undertaken in this work, though, so it is possible that some bacteria not detected by CFU/mL were present. In the future, complementary methods should be employed to ensure complete characterization of bacterial viability.

Hu et al. previously demonstrated the functionality of a CPP and that antisense *motA* PNA treatment reduced biofilm formation in PAO1 in a dose-dependent manner [11]. However, among the probe sequences Hu et al. assessed, the impact of antisense-PNA treatment on bacterial viability was not reported, although decreased motility upon antisense-PNA treatment was observed [11]. In the present study, we demonstrated elimination of biofilm formation and a significant reduction in average bacterial viability upon antisense-PNA *motA* 0 treatment (Figure 1c,d). Interrogating the mechanisms behind how the stator complex of the flagellar motor relates to viability is beyond the scope of the present work; however, future studies could be aimed at characterizing gene and protein expression changes upon *motA* treatment.

To the best of our knowledge, this is the first time antisense-PNAs have been designed to intentionally encompass sequences of genes that are conserved among multiple bacterial species (Table 1). Further, we assessed the impact of these antisense-PNAs on biofilm biomass and bacterial viability (Figure 2). This broadening of PNA specificity is best exemplified by the antisense-PNA *rsmA* 1, whose sequence is applicable to *Enterobacter* spp., *Klebsiella* spp., and *E. coli* (Table 1). Given historical findings that individual antisense-PNA treatment in PAO1 resulted in inconsistent effects (Figure A2), it was not entirely surprising that antisense-PNA treatment did not reduce bacterial viability in the intended species (Figure 2). Future work will be directed at investigating combination therapy that includes both antisense-PNAs and conventional antibiotics to generate a bactericidal effect. However, it was reassuring that biofilm biomass was reduced with antisense-PNA treatment, as the gene targets are global regulators of biofilm formation and maintenance, not necessarily viability (Figure 2). Additional studies could examine differences at the genomic and proteomic levels regarding the discordance between the anti-biofilm effect among the *E. coli* clinical isolates (Figure 2e). Identification of novel, shared genes associated with prominent biofilm-forming phenotypes could offer new targets for antisense-PNA design. Antisense-PNAs targeting four common Gram-negative bacilli were included in the present study; however, expansion of this work could include evaluating off-target anti-biofilm effects and a larger scope of inquiry to include other bacteria, including Gram-negative and Gram-positive bacteria implicated in CAUTIs.

Given that polymicrobial colonization is common among catheterized patients, ranging from 31% to 87% depending on the study, polymicrobial samples were investigated in the laboratory [31]. TSA and MAC agar plates were sufficient for differentiating the Gram-negative bacilli in this study based on phenotypic differences (Figure 3a). The loss of the anti-biofilm effect of antisense-PNA *rsmA* 1 when combined in an equimolar cocktail with antisense-PNAs *amrZ* 0 and *rpoS* 0 was unexpected (Figure 3b). Binding among the antisense-PNAs likely did not occur, as PAO1 viability was still eliminated with the *rsmA* 1, *amrZ* 0, and *rpoS* 0 cocktail (Figure 3c). Instead, downstream effects from the cPNA may be responsible, which requires further investigation at the genomic and proteomic levels. The consistent reduction in PAO1 bacterial viability upon treatment suggests that the two base pair difference in the sequence was not sufficient to prevent an effect from the antisense-PNA designed for *Enterobacter* spp., *Klebsiella* spp., and *E. coli* on PAO1 (Figure 3c and Table 1). The PNA literature describes the peptide-like backbone of antisense-PNAs as enhancing specificity, though the effect that antisense-PNAs have on specificity requires additional study [15]. Future work could be directed at characterizing the individual contribution of different bacterial species to overall biofilm biomass and assessing other combinations, or more complex combinations, of polymicrobial samples. Additionally, future studies could evaluate the extent of PNA integration by quantifying the target gene expression.

Ultimately, antisense-PNA technology holds potential for a range of applications, including as a coating on the luminal surface of catheters to prevent biofilm formation and bacterial viability. Drying the cPNA onto the surface of 96-well plates was a preliminary trial in increasing clinical applicability, and consistent biofilm formation between the untreated conditions supported this methodology (Figure 4). Further, the aqueous and dried cPNA conditions both completely eliminated biofilm biomass and bacterial viability of *Pa383-*∆*rahU*::GM, suggesting that the return to a lyophilized state did not inherently eliminate the ability of the cPNA to penetrate bacteria, hybridize to bacterial DNA, and exert their intended bioactivity (Figure 4). Given that PNAs may be suitable for intraluminal coating and are customizable in nature, one could envision a way that PNAs could be designed to promote healthier microbiota and deflect pathogenic species from catheter surfaces.

Antisense-PNAs could be used to prevent biofilm-associated infections in other contexts, including surgical meshes, burn wounds, pancreatic and biliary stents, and implantable medical devices [32]. This approach is particularly relevant for patients with long-term urinary catheters, such as for urinary retention, bladder outlet obstruction, and pelvic fracture. For catheters colonized with polymicrobial bacteria, antisense-PNAs capitalize on genetic conservation among commonly implicated bacterial species in CAUTIs. In addition, antimicrobials could be used in combination with effective antisense-PNAs in settings of chronic catheterization. Thus, antisense peptide nucleic acids offer a promising solution to many clinically relevant problems.

## 5. Conclusions

This study supports the use of novel antisense peptide nucleic acids to target the irreversible and reversible phases of biofilm formation to reduce or fully eliminate biofilm biomass by common Gram-negative bacilli implicated in catheter-associated urinary tract infections. Although a cocktail of peptide nucleic acids demonstrated a bactericidal effect against a biofilm-forming laboratory strain of *Pseudomonas aeruginosa*, the other novel peptide nucleic acids evaluated in this study only demonstrated anti-biofilm effects against their intended species.

## 6. Patents

The non-provisional patent application entitled “Wide Range Synthetic Peptide Nucleic Acids and Their Applications” was filed on 7 May 2025 (United States application number: 19/201,032).

## Figures and Tables

**Figure 1 microorganisms-13-01948-f001:**
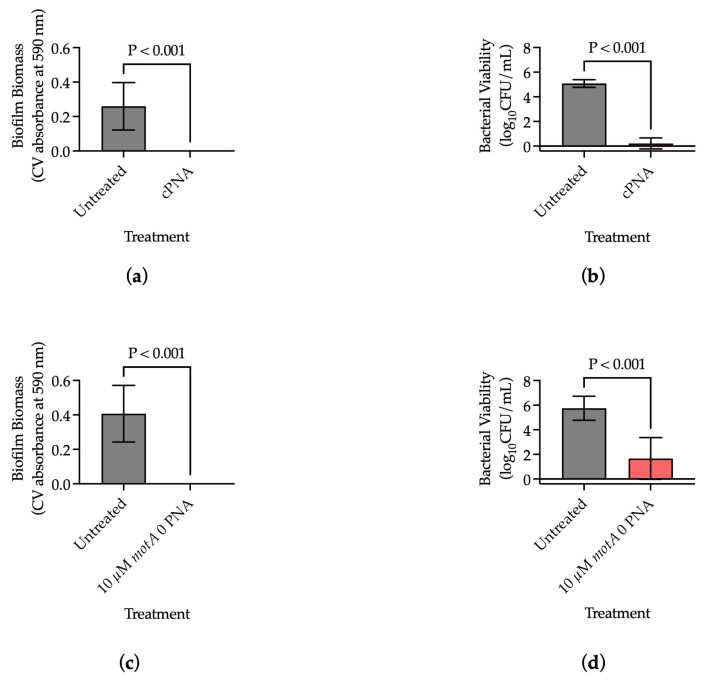
Antisense-PNAs reduced biofilm formation and bacterial viability among PAO1. (**a**) Biofilm biomass after 24 h cPNA treatment (n = 15); (**b**) Bacterial viability after 24 h cPNA treatment (n = 15); (**c**) Biofilm biomass after 24 h treatment with antisense-PNA *motA* 0 (n = 9); (**d**) Bacterial viability after 24 h treatment with antisense-PNA *motA* 0 (n = 9). Results are presented as mean ± s.d., and *p* values were determined using unpaired *t*-tests (α = 0.05).

**Figure 2 microorganisms-13-01948-f002:**
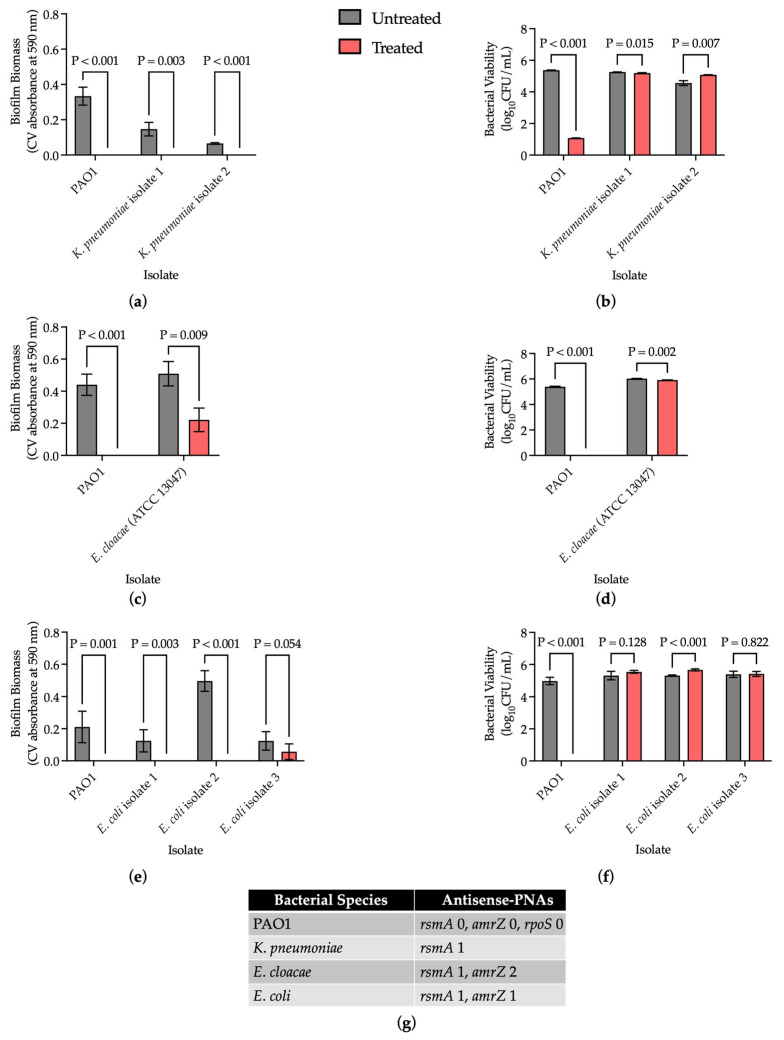
Antisense-PNA treatment reduced biofilm biomass in their intended bacterial species with no meaningful reduction in bacterial viability. (**a**) Biofilm biomass after 24 h antisense-PNA treatment in *K. pneumoniae* clinical isolates; (**b**) Bacterial viability after 24 h antisense-PNA treatment in *K. pneumoniae* clinical isolates; (**c**) Biofilm biomass after 24 h antisense-PNA treatment in commercial *E. cloacae* strain ATCC 13047; (**d**) Bacterial viability after 24 h antisense-PNA treatment in ATCC 13047; (**e**) Biofilm biomass after 24 h antisense-PNA treatment in *E. coli* clinical isolates; (**f**) Bacterial viability after 24 h antisense-PNA treatment in *E. coli* clinical isolates; (**g**) Summary of the antisense-PNAs utilized for each bacterial species. PAO1 was treated with cPNA. Results are presented as mean ± s.d., and *p* values were determined using unpaired *t*-tests (α = 0.05) (n = 3; except for (**e**) and (**f**), n = 6).

**Figure 3 microorganisms-13-01948-f003:**
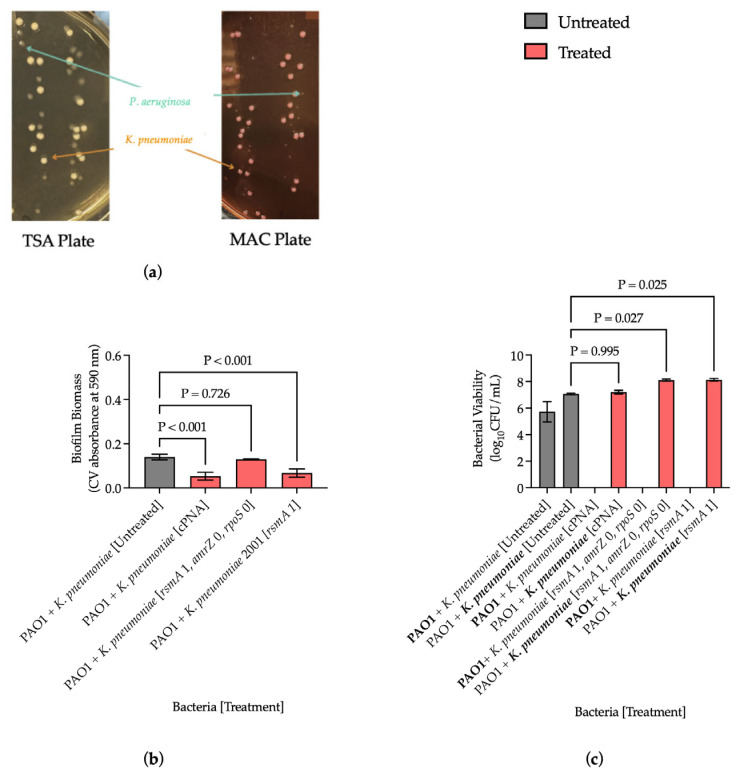
Antisense-PNAs reduced biofilm biomass in a polymicrobial sample of 1:1 PAO1:*K. pneumoniae*. (**a**) The mucoid phenotype of *K. pneumoniae* is highlighted on MacConkey (MAC) agar plates compared to Tryptic Soy Agar (TSA) plates; (**b**) Biofilm biomass after 24 h antisense-PNA treatment; (**c**) Bacterial viability after 24 h antisense-PNA treatment. The bolded bacterial species is the component for which the bacterial viability is represented. All conditions were tested in triplicate. Results are presented as mean ± s.d., and *p* values were determined using one-way ANOVA with follow-up tests comparing the mean of each column to the control column (α = 0.05).

**Figure 4 microorganisms-13-01948-f004:**
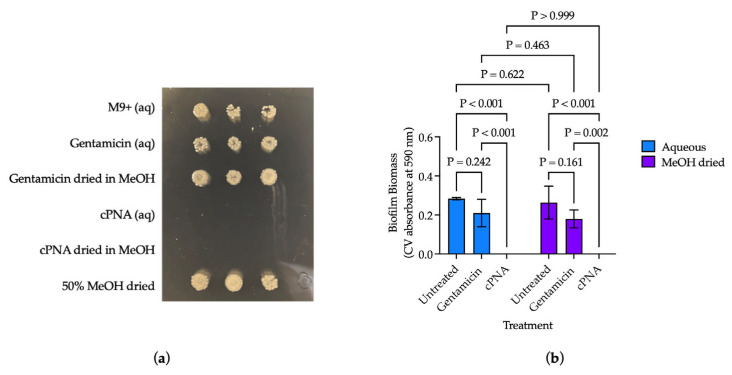
Dried cPNA retained their anti-biofilm effect against gentamicin-resistant *P. aeruginosa*, *Pa383*-∆*rahU*::GM. (**a**) Qualitative viability after 24 h incubation; (**b**) Biofilm biomass after 32 h cPNA treatment in aqueous (aq) solution or dried onto the 96-well plate with methanol (MeOH). All conditions were tested in triplicate. Results are presented as mean ± s.d., and *p* values were determined using two-way ANOVA with multiple comparisons comparing cell means with others in its row and its column (family-wise α threshold and confidence level = 0.05).

**Table 1 microorganisms-13-01948-t001:** Design of antisense peptide nucleic acids (PNAs). The start codon (ATG) is highlighted in yellow. The intended bacterial species for each antisense-PNA is in parentheses.

Antisense-PNA	Sequence													
*rsmA* 0 (*P. aeruginosa*)	G	A	A	A	G	G	A	A	T	G	C	T		
*rsmA* 1 (*Klebsiella* spp., *Enterobacter* spp., and *E. coli*)	G	C	A	A	A	G	A	A	T	G	C	T		
*amrZ* 0 (*P. aeruginosa*)	A	A	T	G	T	A	T	G	C	G	C	C		
*amrZ* 1 (*E. coli*)	G	T	C	A	T	A	T	G	A	G	C	A		
*amrZ* 2 (*Enterobacter* spp.)	G	C	G	C	C	A	T	G	A	C	G	A		
*rpoS* 0 (*P. aeruginosa*)	G	G	A	T	A	A	C	G	A	C	A	T	G	
*motA* 0 (*P. aeruginosa*)	C	C	T	C	A	T	G	T	C	A	A	A	A	A

## Data Availability

The raw data supporting the conclusions of this article will be made available by the authors on request.

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
