# Peer review of "Broad Host Range Peptide Nucleic Acids Prevent Gram-Negative Biofilms Implicated in Catheter-Associated Urinary Tract Infections"

_microorganisms, 2025, doi:10.3390/microorganisms13081948_

Round 1

Reviewer 1 Report

Comments and Suggestions for Authors

Karp et al. presented a study on the use of PNAs to prevent biofilm growth on urinary catheters, demonstrating the potential of this antisense technique for preventing catheter-associated urinary tract infections (CAUTI). While the insights provided are highly relevant and valuable, the review raised several questions, particularly concerning some experimental details that require further clarification. Addressing these aspects would enhance the overall understanding of the study and its implications for future research.

- The title of the manuscript is somewhat misleading, as it implies that this tool is ready for clinical application. While the study introduces a promising new approach for preventing Gram-negative biofilms, it is not yet fully established for clinical use. The authors should revise the title to more accurately reflect the current stage of the research.

-Line 70: The manuscript would benefit from a more detailed definition and explanation of antisense-wide-range PNAs (wrPNAs). While the term is introduced, the authors do not provide sufficient context or clarification on how wrPNAs differ from antisense PNAs. A more thorough description of specific advantages in the context of the study would enhance the  understanding of the experimental design.

-Taking from the previous point, the manuscript would also benefit from clearer differentiation between antisense PNAs and wrPNAs, particularly regarding when and why each was used. For example in Table 1, which is titled "antisense-wrPNA," some probes with a “0” for P. aeruginosa appear to correspond to standard antisense PNAs rather than wrPNAs. This creates some confusion in interpreting the results. A more detailed explanation of the criteria for selecting each type of probe, along with consistent and accurate labeling throughout the table and figures, would improve clarity and help to better understand the experimental flow.

-2.1 Bacterial strains: More information on the different strains used is needed.

- Line 92:  More information on the cell-penetrating peptides (CPPs) employed and how they were conjugated to the PNAs could improve the quality of the manuscript. Specifically, a clearer explanation of the conjugation methodology would be helpful, as this could potentially explain why some of the probes are not functioning as expected. It also lacks clarity regarding the transfection process used for the different bacterial strains. A more detailed description of this process, along with any associated experiments, would significantly strengthen the study's findings and enable the results to be interpreted more easily.

-2.4. Drying a cocktail of PNAs onto 96-well plates. The methodology requires further clarification. The authors mention the preparation of cPNAs for antisense PNAs, but it is unclear whether combinations with wrPNAs were also utilized, as it is implied in the results. This aspect should be clearly addressed in the methods section to avoid confusion and ensure that the practical work is fully understood.

-2.5. Bacterial viability assessments: While the results appear unaffected by the control method, there are concerns regarding the potential presence of viable but non-culturable cells, which may not be detected by CFUs. The authors should address how this potential limitation was considered and whether alternative or complementary methods were employed to assess bacterial viability more comprehensively.

-Table 1: It is unclear why the probes targeting the rpoS and motA genes do not have corresponding wrPNAs and why these probes were not tested against other Gram-negative bacteria. It would be helpful for the authors to provide a rationale for these decisions and discuss whether testing with other Gram-negative bacteria could offer further insights or strengthen the findings of the study.

-Table 1: It was not possible to distinguish the bold, but this detail may be related to the computer settings. Just a warning to make it easier to distinguish.

-Figure 2: For improved clarity and ease of interpretation, the corresponding bacterial strains should be labelled more clearly in each graphic. Additionally, it would be beneficial to include a table summarizing the combinations of probes used in each assay, as this would provide a quick reference and enhance the overall readability of the results.

-Figure 3: The authors argue that morphology alone is sufficient to distinguish between the two bacterial strains. However, in panel B, the potential superimposition of one strain onto the other appears to make this distinction unclear. Please provide additional evidence to ensure that the morphological differences are truly sufficient for accurate identification.

Reviewer 2 Report

Comments and Suggestions for Authors

The authors present and interesting study on a novel approach for the prevention of clinical biofilms using artificial peptide-based antisense nucleotides. I have just a few questions and comments that could use addressing.

Line 98: Does this refer to overnight (and thus stationary phase) cultures that have been diluted to a 0.01 OD equivalent? It is very helpful that comparable cell numbers were confirmed with plate counts.
Line 100: 8-hydroxyquinolone? Is this supposed to be hydroxyquinoline, and what is its function in this medium?
Line 168: A reminder that dead bacteria do not produce anything including biofilm, therefore the lack of biofilm in Fig. 1a is not strong evidence of direct cPNA anti-biofilm action.
Line 278: This paragraph does not seem to adequately address why motA PNA was inhibitory to viability but a similar PNA in a previous study was not. The answer is not in Table 1 which does not list the sequence from the previous study, nor is there an explanation why inhibition of motility would result in cell death.
Line 287: On the one hand, improved bactericidal effect may be a plus, but a major advantage of a targeted, anti-biofilm effect is the prevention of collateral damage to the host microbiota. It could be argued that optimization of the range and efficiency of anti-biofilm action should be the goal.
